# The Impact of Charging Infrastructure on Local Emissions of Nitrogen Oxides

## Karsten Hager * and Alexandra Graf

Institut Stadt | Mobilität | Energie (ISME) GmbH, Rotenwaldstraße 18, 70197 Stuttgart, Germany
* Correspondence: karsten.hager@i-sme.de

**Abstract:** Benefits from EV (Electric vehicles) and e-mobility include the reduction of local emissions of pollutants from particulate matter ($PM_{0.5}$, $PM_5$, and $PM_{10}$) and nitrogen oxides ($NO_x$ and $NO_2$). Cities and urban agglomerations benefit the most from potential emission reductions from EVs due to the large number of cars utilized in most urban traffic systems. This abstract presents results from a corporate research and funding project in Baden-Wuerttemberg, Germany (LINOx BW) which facilitates the installation of 2358 charging points within 178 different sub-projects in 23 different cities, spanning a period of four years. Utilizing several different survey waves, data about outgoing currents from these publicly funded charging points are gathered. Converting this data utilizing car classifications and emission classes (HBEFA), the reduction of local nitrogen oxides is derived.

**Keywords:** BEV (battery electric vehicle); charging; infrastructure; pollution; research; nitrogen oxides



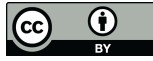

## 1. Introduction

Limit values for nitrogen oxide emissions were adopted by the European Commission in 2008 and are mandatory for all member countries since 2010 [1]. Legal measures regarding nitrogen oxide emissions were not only taken in Europe but, for example, in the U.S. as well [2]. It is well known that nitrogen oxides have significant negative impacts on human health: the fundamentals of the influence of nitrogen oxides on public health concerns are shown in [3]: increasing the risk of respiratory tract infections [4], increasing the incidence and exacerbation of respiratory disease [5], comparing nitrogen oxides to other air pollutants [6], and case studies from New Zealand [7], China [8] and Malaysia [9]. This is especially the case for transport-related pollution, increasing the risk of death from cardiopulmonary causes [10], increasing the risk of childhood asthma [11], and asthma prevalence [12]. BEV and PHEV lead to positive health and air-quality co-benefits [13] even though there are also natural sources of nitrogen oxide emissions [14].

By 2017, 65 different cities in Germany measured nitrogen oxide values far greater than the threshold of 40 $\mu g/m^3$ on a daily basis [15]. Cities exceeding this threshold for a certain amount of time are legally responsible to adopt measures to reduce nitrogen oxide emissions; e.g., driving bans for diesel cars in city centers. The quantification of positive environmental effects such as pollution reduction of charging infrastructure has not yet been sufficiently addressed by research. Addressing this problem, the German Federal Ministry for Economic Affairs and Climate Action (BMWK) released a funding program facilitating the construction of charging infrastructure for the impacted cities in 2017 (Sofortprogramm "Saubere Luft 2017–2020"). Quantifying the reduction of nitrogen oxide emissions from charging infrastructure throughout the research program is an ongoing scientific task. Previous studies addressing this problem utilized car tracking devices combined with real-world driving scenarios [16], applied spatial panel data approaches [17], or executed well-to-wheel (WTW) approaches [18]. The importance of low-emission technologies and their essential role in the future is highlighted by materials utilized for batteries [19], energy savings from Plug-in Hybrid electric vehicles (PHEV) [20], and the utilization and comparison of fuel-cell electric vehicles (FCEV) and BEV [21].

Deriving potential emission reductions from charging infrastructure having no primary link between each other is the novelty of the presented approach. Two research concepts are presented utilizing different resolutions of collected data leading to potential transferable results from other research projects. Discussion of the presented approach is just the first step towards additional required research quantifying the emission reduction potential of charging infrastructure.

The corporate research project "Ladeinfrastruktur in NOx-Kommunen in Baden-Württemberg" (LINOx BW) consists of 178 recipients from 23 different cities in Baden-Württemberg [22]. Grant applications from interested proposers inhabiting the impacted cities in Baden-Württemberg were bundled within the research project to simplify administrative processes. Additionally, accompanying research for all recipients is conducted consistently by two research institutes. So far, 2358 charging points were funded within the research project, utilizing around 10 million euros of federal subsidies. Figure 1 presents the amount of charging points within each city.

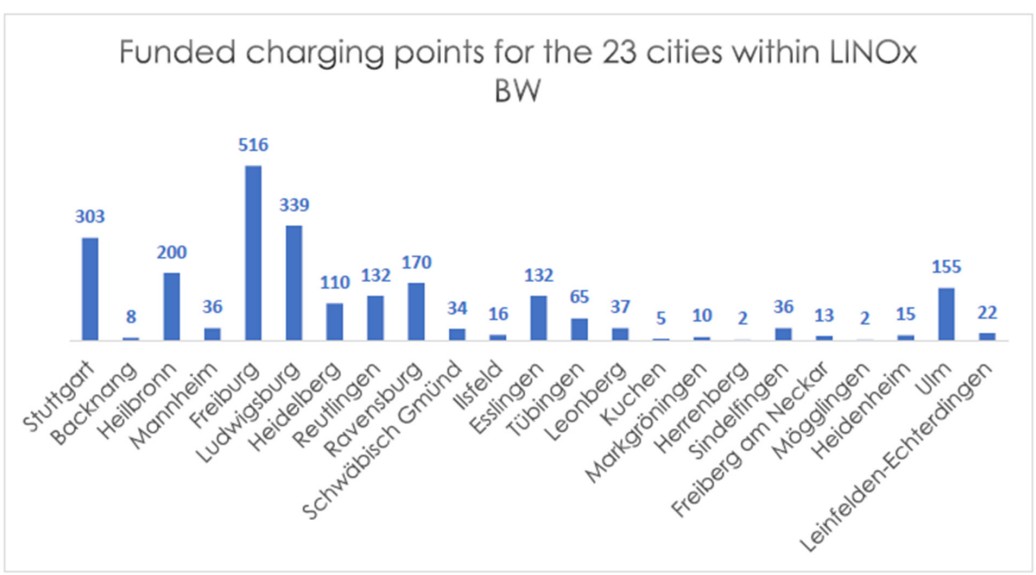

**Figure 1.** Funded charging points for the 23 cities within LINOx BW.

## 2. Materials and Methods

Charging infrastructure projects were categorized into seven different use cases by researchers based on potential EV driving patterns (and therefore different charging opportunities) or different spatial boundary conditions: residential quarter, semi-public space, parking lots and park and ride, private on-site parking, nursing services, tourism, and e-bikes (pedelecs). Utilizing a mixed methods approach consisting of quantitative and qualitative surveys, all recipients received a $t_0$ survey at the start of their respective project about their status quo containing questions, among others, relating to their vehicle fleets, existing charging infrastructure, parking lot size, and enterprise size. A different questionnaire was set up including specific questions for each use case. The goal of the research project is to gather as much information about user experiences, vehicle fleets, and the amount of electricity delivered as possible, thus, at the end of their respective project, funding recipients received a $t_1$ survey containing the same questions as the $t_0$ survey. This procedure allows for a comparison between a point in time before the construction of the charging infrastructure and a point in time afterward.

Additionally, hand-picked recipients planning exceptionally large or special charging infrastructure projects were chosen for a one-hour telephone interview and are referred to as lead partners. The utilized semi-structured interview guideline comprises four main sections: decision-making and communication procedures, cost, revenue, and operator models, use intensity of charging infrastructure, and use-case-specific aspects. These use-

case-specific questions allow for the accommodation of use cases such as nursing services or tourism, which are very different from private on-site parking, the use case that is encountered most often in the funding program.

Presented results in the paper refer to data from the cutoff date of 15 January 2023. The official project end is set for 31 December 2023, so funding recipients receive sufficient time for their respective depreciation periods caused by, e.g., delays from delivery bottlenecks for charging infrastructure. New funding recipients will not be approved beyond this point. The research presented in this paper is based on the evaluation of 82 questionnaires—82 funding recipients finished the realization of their respective projects and depreciated their charging infrastructure. Referring to the introduction, data from almost 100 recipients of LINOx BW could not be utilized within this paper because their projects are unfinished as of now. The first project results for parking lots and private areas were already published in German [23].

One of the goals of LINOx BW is to quantify the reduction of local nitrogen oxides when charging infrastructure is built. Therefore, the amount of electricity delivered by the funded charging points is the key variable to quantify the impact of charging infrastructure on local emissions of nitrogen oxides—overall and within the project. Utilizing these values, the amount of electricity is converted to electric vehicle kilometers based on average consumption values of kWh/100 km for different car classifications from electric car databases. From the collected data, two results are derived:

(1) The current $NO_x$-reduction based on the amount of electricity of the funded charging infrastructure divided by the average consumption value. For this paper, 18.4 kWh/100 km is utilized, based on literature [24] (p. 17). To derive the amount of NOx reduction, an average $NO_x$ factor (g/km) based on the average German fleet composition (2022) included in the current version of HBEFA and the HBEFA traffic scenario "D Ø UBA 2021" is utilized. This approach is utilized for all surveys containing data on delivered electricity; no fleet of vehicle data is required to derive nitrogen oxide reductions.

(2) The maximum $NO_x$-reduction based on a 1:1 substitution for all cars in the respective vehicle fleets utilizing a fleet of vehicle data (mileage and car classifications) to calculate NOx-reduction based on all cars in the respective vehicle fleets utilizing emission factors for ICEV based on the Handbook Emission Factors for Road Transport (HBEFA, [25]). In research concept (2) surveys where car emission concepts are unknown, average emission factors are calculated based on HBEFA and utilized for the calculation of the $NO_x$-reduction values. Figure 2 summarizes the research concept for this paper; Tables 1 and 2 show the utilized passenger car and heavy-duty vehicle emissions concepts and their respective HBEFA emissions factors utilized for this research.

**Table 1.** Passenger car emission concepts and HBEFA emission factors.

| Emission Concept | Share (%) in German Fleet Composition | HBEFA-Value (g/km) |
|---|---|---|
| petrol (4S) | 51.08% | 0.091258764 |
| diesel | 45.22% | 0.599183857 |
| electricity | 0.98% | 0 |
| bi-fuel CNG/petrol | 0.25% | 0.170386568 |
| plug-in hybrid petrol/electric | 1.50% | 0.021050557 |
| plug-in hybrid diesel/electric | 0.17% | 0.085323997 |
| bi-fuel LPG/petrol | 0.80% | 0.172697365 |

**Table 2.** Heavy-duty vehicle emission concepts and HBEFA emission factors.

| Emission Concept | Share (%) in German Fleet Composition | HBEFA-Value (g/km) |
|---|---|---|
| petrol (4S) | 4.66% | 0.14836058 |
| diesel | 94.12% | 0.816266835 |
| electricity | 0.79% | 0 |
| bi-fuel CNG/petrol | 0.36% | 0.066065677 |
| plug-in hybrid petrol/electric | 0.06% | 0.013855861 |

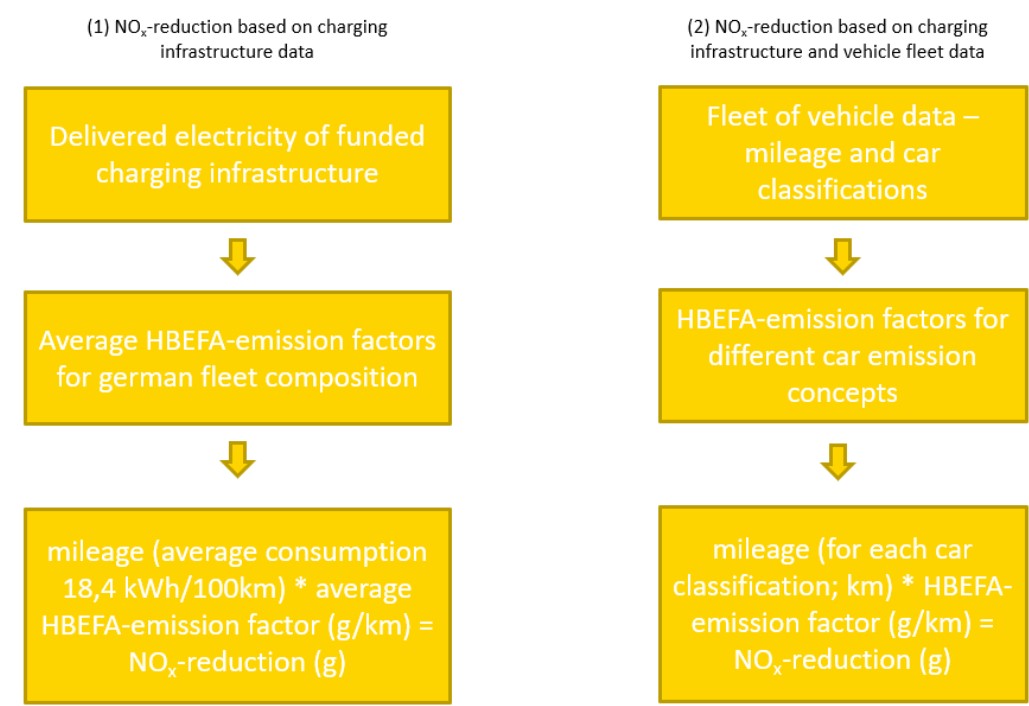

**Figure 2.** Research concept of LINOx BW for classifying the impact of charging infrastructure on local emissions of nitrogen oxides.

The average HBEFA value for surveys lacking fleet data is 0.371688628 g/km (based on a share of passenger cars and heavy-duty vehicles in German fleet composition, taken from [26]). For cases containing passenger cars with unknown emission concepts, an average value utilizing the data from Table 1 was calculated: 0.319849483 g/km. Similarly, for heavy-duty vehicles, the average value is 0.775203772 g/km. Euro certificates of cars could not be utilized for the research, because these data could not be provided by the funding recipients, even though the HBEFA provides different emissions factors related to Euro certificates.

Unfortunately, independent scientific tracking of all funded charging points could not be provided within the research project due to funding guidelines; thus, research is dependent on survey data from funding recipients measuring and documenting the amount of electricity delivered by charging infrastructure. Many funding recipients were small- and medium-sized companies, and the quality of completed surveys differed greatly between recipients with familiarity with research projects as well as familiarity with the field of electric mobility. However, the results represent many different use cases containing data from daily charging and BEV utilization routines potentially leading to realistic results where data are provided. The driving patterns of the respective vehicle fleets could not be provided within the research project. The completion of the $t_1$ survey, however, was mandatory for all funding recipients. Without approval from the research group that the survey has been completed, the last 10% of funding money would not be disbursed.

## 3. Results

### 3.1. General Results

An overview of the collected data from the $t_1$ survey is shown in Table 3. Additionally, Figure 3 shows the spatial location of the 82 questionnaires within the LINOx BW cities. Questionnaires hail from 17 of the 23 possible cities with the vast majority located in the city of Freiburg (25 or 30.4% of all 82 questionnaires), followed by the city of Stuttgart (11 or 13.4%) and the city of Ulm (9 or 11%).

**Table 3.** Key figures about utilized data for this paper.

| Number of | Amount |
|---|---|
| Questionnaires | 82 |
| Surveys with charged electricity data | 63 |
| Surveys with a fleet of vehicle data | 33 |
| Charging points before LINOx BW | 131 |
| Charging points with/after LINOx BW | 949 |
| BEV purchased since funding of charging infrastructure through LINOx BW | 124 |
| PHEV (diesel) purchased since funding of charging infrastructure through LINOx BW | 25 |
| PHEV (petrol) purchased since funding of charging infrastructure through LINOx BW | 94 |
| Pedelecs purchased since start of LINOx BW | 61 |
| Surveys replacing passenger cars with pedelecs | 5 |
| Passenger cars in existing vehicle fleets | 1615 |
| Passenger cars utilizing diesel | 739 |
| With unknown car emission concept | 248 |
| Heavy-duty vehicles in existing vehicle fleets | 486 |
| Heavy-duty vehicles utilizing diesel | 190 |
| With unknown car emission concept | 236 |
| Installed charging points (AC) | 799 |
| Installed charging points (DC) | 19 |
| Sum of installed charging infrastructure power (AC) (kW) | 12,899.5 |
| Sum of installed charging infrastructure power (DC) (kW) | 1365 |
| Charging capacity/charging point (kW) | 17.4 |

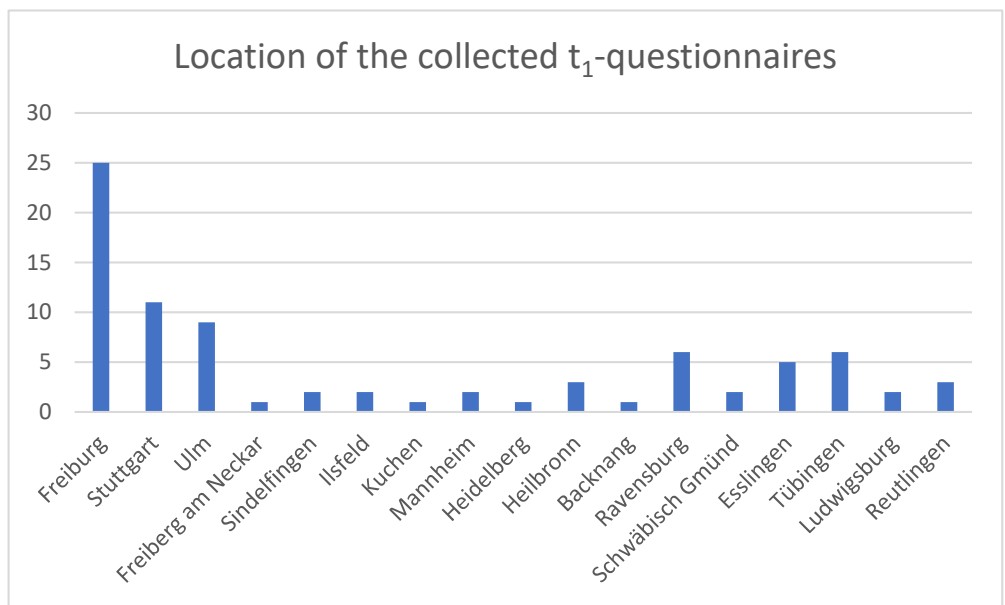

**Figure 3.** Spatial location of the collected $t_1$-questionnaires within LINOx BW cities.

Out of the 82 questionnaires collected, only 63 collected charged electricity data from the funded charging infrastructure. Reasons for 19 cases without these data are, e.g., no separate electric meters for charging infrastructure, thus collected data would include household electricity as well. A total of 63 cases were utilized for research concept (1); see Section 3.2. All questionnaires containing vehicle fleet data, 33 to be precise, were utilized for research concept (2); see Section 3.3.

During the duration of the project, 818 charging points were installed throughout the 82 questionnaires utilized for this paper, corresponding to 34.6% of the charging points funded within the whole project. Results shown in this paper consist of approximately one-third of the overall project results.

Vehicle fleets contain an additional 124 BEV, 25 PHEV (diesel), and 94 PHEV (petrol) since the start of the project—to summarize, 243 electric vehicles for 818 charging points

(29.7%) relate to additional electric vehicles from visitors, employees, etc., utilizing the charging infrastructure. This highlights future potential for additional electric cars in their respective fleet of vehicles for BEV based on their higher absolute electric range compared to PHEV. In total, survey data shows 303 electric vehicles utilized at the time of data collection within the examined vehicle fleets. During the project period, 243 out of 303 electric vehicles were purchased by the funding recipients (80.2%).

Funding recipients were also allowed to install charging infrastructure for e-bikes (pedelecs). A total of 61 pedelecs were purchased within the project period, with five recipients additionally replacing ICV with pedelecs. No independent tracking of delivered electricity for pedelecs occurred. Quantifying the success and utilization of pedelecs is not possible with the available survey data.

The overall vehicle fleets contain 1615 passenger cars and 486 heavy-duty vehicles (2101 vehicles in total); 243 electric vehicles correspond to 11.5% of all vehicles in the presented vehicle fleets. Unfortunately, 248 passenger cars (15.4%) and 236 heavy-duty vehicles (48.6%) could not be assigned to a car emission concept from HBEFA leading to the increased utilization of the average HBEFA values as shown in Section 2 and thus potential inaccuracies. The largest share of emission concepts for either passenger cars (45.8%) or heavy-duty vehicles (39.1%) within the survey data is diesel. Heavy-duty-vehicle data unfortunately have a higher share of unknown emission concepts than the respective largest share of emission concepts in diesel leading to potential inaccuracies utilizing this data for further calculations.

In total, the sum of the installed charging points and charging infrastructure power for alternate current (AC) charging infrastructure sums up to 799 points and 12,899.5 kW; for direct current (DC), charging infrastructure to 19 points 1365 kW were put into operation during LINOx BW combining for 611 points and 14,264.5 kW in total. Dividing the charging capacity through the number of installed charging points, a factor of 17.4 kW/charging point is derived.

### 3.2. Nitrogen Oxide Reduction Research Concept (1)

Table 4 shows key figures referring to the nitrogen oxide reduction within LINOx BW. The number of valid cases for the reduction calculation from research concept (1) is 63. These 63 cases represent 611 funded charging points (74.7%) and specified values of delivered electricity from the funded charging infrastructure summarizing to 1,600,045.64 kWh. Deriving the approximate electric mileage utilizing 18.4 kWh/100 km values, 8,695,900.22 km will be utilized for calculating the approximate nitrogen oxide reduction. Combining this value with the average HBEFA value for all surveys without vehicle fleet data, approximately 3.2 tons of nitrogen oxides were avoided within the research project.

**Table 4.** Key figures for research concepts (1) and (2).

| Name | Value |
|---|---|
| Number of valid cases for research concept (1) | 63 |
| Delivered electricity from funded charging infrastructure (sum; kWh) (1) | 1,600,045.64 |
| Electric mileage utilizing (18.4 kWh/100 km) (km) (1) | 8,695,900.22 |
| $NO_x$-reduction (sum; g) (1) | 3,232,167.22 |
| Number of valid cases for research concept (2) | 27 |
| Electric mileage utilizing fleet of vehicle data (km) (2) | 9,854,093.28 |
| $NO_x$-reduction (sum; g) (2) | 35,247,610.06 |
| Difference between electric mileage research concept (1) and (2) | −1,158,193.06 |

### 3.3. Nitrogen Oxide Reduction Research Concept (2)

Only 33 questionnaires contained vehicle fleet data—other questionnaires either do not possess a vehicle fleet or could only provide insufficient data regarding their vehicle fleet; e.g., a few respondents could not provide start-up dates for their respective charging infrastructure. The lack of this information is problematic since the data had to be distributed evenly over

the survey period. This results from the fact that not the total kilometers but the kilometers per year were surveyed. Additionally, several valid cases were able to provide vehicle fleet data, but no charged electricity data. Thus, only 27 valid cases were utilized for the reduction calculation for the research concept (2). The valid cases represent only 117 charging points (14.3%). Added-up mileage data for all BEV, PHEV (diesel), and PHEV (petrol) is equivalent to 9,854,093.28 km. To derive a potential nitrogen oxide reduction, HBEFA values from Tables 1 and 2 for each respective vehicle fleet are utilized. In total, roughly 35.2 tons of nitrogen oxides were avoided within the research project.

Comparing the electric mileage from each research concept, however, a difference of 1,158,193.06 km is calculated, meaning the funded charging infrastructure of LINOx BW is not covering the electric mileage of the vehicle fleet data surveys. Even though comparing results from 63 valid cases to 27 valid cases, the important fact is that the difference is negative. A few possible reasons can be considered to explain this:

- Semi-public accessibility in several cases, leading to external electric cars utilizing the charging infrastructure (tourists, visitors, etc.);
- Private utilization of electric cars belonging to a company's vehicle fleet;
- Charging infrastructure owner's permission for employees to utilize the charging infrastructure for their private cars;
- Unknown delivered electricity values due to no data collected or no separate electric meters for the funded charging infrastructure;
- Unknown vehicle fleet data or unusual mileage values;
- The special case of the installation of charging infrastructure in homeowners' associations—no vehicle fleet data and no charged electricity are provided, even though dozens of charging points are installed.

*3.4. Comparison of Both Research Concepts Based on the Valid Cases of Research Concept (2)*

Table 5 presents more detailed information about the valid cases of research concept (2). Here, the direct comparison between the potential NOx-reduction based on the different research concepts is shown. Few valid cases provide vehicle fleet data, thus valid for research concept (2), but did not provide charged electricity data, thus invalid for research concept (1).

First of all, the number of charging points installed before funding of LINOx BW has to be compared to the number of newly installed charging points with project funding. A total of 22 cases installed their first charging infrastructure within the project; thus, they started to utilize electric mobility. The other five cases already had charging points installed, leading to potentially higher values from research concept (2). In most cases, the larger the vehicle fleet provided in the questionnaire, the higher the result for research concept (2). Number 82 is a huge company with several thousands of employees, a vehicle fleet of 511 passenger cars, and a lot of charging infrastructure; however, only four charging points were funded within LINOx BW, questioning the eligibility of outliers for research concept (2). Without number 82, the potential nitrogen oxide reduction for research concept (2) results in only 2.05 t.

The variable days of data acquisition are irrelevant to research concept (1) and can potentially be neglected for future research utilizing research concept (1).

Additionally, a few municipalities completed the survey providing information about the vehicle fleet of the whole municipality—not only for the site their charging infrastructure was installed. Unknown car emission concepts need to be calculated with average HBEFA values, which are considerably higher especially comparing diesel and petrol cars. Summarizing the comparison, Table 5 shows the data heterogeneity based on the survey data complicating the task of appropriate research concepts to analyze the project's data.

The utilization of additional descriptive factors, e.g., daily charged electricity was neglected due to the fact that most funding recipients have started utilizing their funded charging points on separate dates and thus daily charged electricity values would need to be averaged. Total values of charged electricity suffice for a preliminary comparative analysis.

**Table 5.** Detailed information about the 27 valid cases for research concept (2).

| Number | Charging Points ex. LINOx BW | Charging Points by LINOx BW | Charged Electricity (kWh) | Days of Data Acquisition of Charged Electricity | NOx-Reduction (1) (g) | Number of Passenger Cars | Number of Heavy-Duty Vehicles | NOx-Reduction (2) (g) |
|---|---|---|---|---|---|---|---|---|
| 1 | 0 | 8 | 8043 | 336 | 16,247.24 | 1 | 1 | 6097.64 |
| 2 | 0 | 5 | 7703 | 559 | 15,560.42 | 1 | 2 | 21,265.52 |
| 3 | 0 | 2 | 6147 | 1008 | 12,417.23 | 6 | 0 | 51,835.74 |
| 10 | 2 | 5 | 6763 | 1501 | 24,969.80 | 5 | 9 | 112,723.54 |
| 14 | 0 | 1 | 3633 | 386 | 7338.83 | 20 | 2 | 70,910.60 |
| 20 | 0 | 4 | No data | No data | 0.00 | 8 | 0 | 57,703.04 |
| 21 | 0 | 5 | 1740 | 314 | 3514.88 | 6 | 0 | 9028.37 |
| 32 | 2 | 21 | 11,298.12 | 765 | 53,161.80 | 28 | 104 | 62,913.07 |
| 36 | 0 | 6 | 18,717 | 719 | 37,809.22 | 89 | 0 | 480,019.94 |
| 41 | 0 | 4 | 12,110 | 541 | 24,462.77 | 18 | 0 | 16,231.56 |
| 46 | 0 | 4 | 1579 | 99 | 3189.65 | 4 | 36 | 78,903.78 |
| 47 | 0 | 7 | 172,515 | 602 | 348,488.39 | 11 | 0 | 123,104.59 |
| 48 | 0 | 2 | No data | No data | 0.00 | 5 | 0 | 52,439.56 |
| 49 | 0 | 6 | 8271 | 423 | 17,998.62 | 45 | 1 | 56,195.56 |
| 59 | 0 | 10 | 14,875 | 700 | 30,048.20 | 28 | 2 | 8535.81 |
| 64 | 0 | 2 | 2018 | 424 | 4076.45 | 3 | 17 | 384,143.37 |
| 68 | 0 | 2 | 4845.22 | 1271 | 9787.57 | 2 | 0 | 21,597.75 |
| 69 | 0 | 3 | 2044.36 | 1628 | 4129.69 | 4 | 1 | 36,037.62 |
| 70 | 0 | 2 | 2580 | 1230 | 5211.72 | 1 | 1 | 420.23 |
| 71 | 0 | 2 | 636.42 | 1257 | 1285.59 | 3 | 1 | 13,976.85 |
| 72 | 4 | 2 | 3654 | 1350 | 7381.25 | 6 | 1 | 88,646.38 |
| 73 | 7 | 1 | 1302 | 1110 | 2630.10 | 9 | 2 | 132,713.05 |
| 74 | 0 | 2 | 896.66 | 341 | 1811.29 | 2 | 1 | 641.09 |
| 75 | 0 | 2 | 1430 | 1319 | 2888.67 | 4 | 1 | 81,244.59 |
| 76 | 0 | 1 | No data | No data | 0.00 | 1 | 0 | 3672.08 |
| 81 | 0 | 4 | 1579 | 99 | 3189.65 | 4 | 36 | 78,903.78 |
| 82 | 87 | 4 | 13,686 | 999 | 27,646.36 | 511 | 0 | 33,197,704.94 |

## 4. Discussion

Contextualizing the presented results, the 818 installed charging points within Baden-Württemberg over the course of the project, a comparison with data from the German Federal Network Agency (Bundesnetzagentur) for installed charging points within Baden-Württemberg follows.

In Germany, charging points with at least 11 kW are subject to registration. By 1 January 2019, the nearest date to our project start with available data, 2180 AC charging points and 437 DC charging points were installed in Baden-Wuerttemberg (total of 2.617 charging points), with a factor of 28.01 kW/charging point. By October 01, 2022, 11,149 AC charging points and 1909 DC charging points were installed in Baden-Württemberg (total of 13.058 charging points), with a factor of 28.03 kW/charging point [27]. During this period, 10,441 charging points have been installed in Baden-Wuerttemberg, with 818 charging points funded by LINOx BW (7.8%). However, the factor of 17.4 kW/charging point within LINOx BW is considerably lower leading to the assumption that LINOx BW facilitated an area-wide roll-out of AC charging infrastructure in the first place. Additionally, small- and medium-sized enterprises as well as considerably small projects with a low amount of charging infrastructure undertaking their first steps towards electric mobility do not require fast charging infrastructure for their respective needs.

The presented results account for only one-third of all possible data collected within LINOx BW. Most of the recipients participating in LINOx BW are unfamiliar with research projects leading to uncertainties when it comes to filling in the survey. Many questions in the questionnaires were not answered and left without remark; moreover, some answers did not correspond to the $t_0$-survey. Additionally, certain data in the respective funding recipient's organization is unavailable due to technical or organizational reasons. The possibility for each recipient to be able to derive electricity data from their funded charging infrastructure could not be made mandatory due to funding guidelines skipping research potential. Recipients that owned several locations but installed charging infrastructure on only one location could not separate their internal data to the locations relevant to LINOx BW. Conducting the results at the end of the project at the end of 2023, standardizing and verifying the collected data, and quantifying the number and the respective amount of side effects such as the allowance for semi-public or employee charging will be key. Additionally, unconventionally large or special cases need to be taken care of differently, as number 82 from Table 5 shows.

Comparing the eligibility of both presented research concepts, (1) represents 74.7% of all funded charging points within current survey data, whereas (2) represents only 14.3%. Providing the required data for research concept (2) is more difficult for the recipients inexperienced with surveys and research projects. Thus, a convergence of both % utilizing the upcoming additional data of not yet finished projects is highly unlikely. Looking toward future research, the utilization of research concept (1) seems more promising.

Final calculations for the emission reduction potential require data from the $t_1$-survey and the respective vehicle fleets. The huge amount of unknown car emission concepts leading to the utilization of average HBEFA factors for the nitrogen oxide reduction potential increases potential inaccuracies for the calculation results. The substitution of ICV to BEV or PHEV differs from recipient to recipient and is also based on other effects such as depreciation periods of currently utilized cars. Quantifying the reduction from recipients with private charging only is rather simple. The utilization of pedelecs adds slightly to the project's success (in particular, if the respective recipients are replacing passenger cars or heavy-duty vehicles); however, quantification is not possible.

In 2020, the German emissions of NOx reached a total of 978,000 tons, with 390,300 tons belonging to the traffic sector [28]. The current gathered LINOx BW data adds up to 35.24 tons (for research concept (2)) and to 3.23 tons (for research concept (1)) and is partly collected from several different years. For scientific comparison, potential emission reductions must be calculated on a yearly basis. Overall, the presented results utilized data from 818 charging points (the whole project funds 2.358 charging points). Effects will

increase with time, leading to the integration of more electric cars into vehicle fleets as well as more electric cars in general in Germany, thus leading to an increased usage from semi-public charging infrastructure. However, all 82 questionnaires presented in this paper will not be surveyed another time, their respective results are final.

## 5. Conclusions

A new concept for quantifying nitrogen oxide emissions from charging infrastructure is presented. The presented approach shows a few limitations based on the project specifications of LINOx BW: No mandatory guidelines for deriving the charged electricity could be implemented, leading to the utilization of assumptions and some cases not being valid for research concepts. The verification of nitrogen oxide values is not possible due to the nature of air pollutants; additionally, no direct access to the required data from the recipients for researchers, thus leading to the utilization of questionnaires.

Gathering the required data for the analysis of nitrogen oxide emission reductions is an uphill task without direct access to the data. The mandatory installation of smart meters and separate electric meters for the funded charging infrastructure would greatly increase the data quality required for the research. Additional information about specific user groups utilizing the funded charging infrastructure would increase the scientific insight as well. The reliability of mileage and a fleet of vehicle data needs to be addressed for future research as well.

The analysis of the results so far has not considered the seven different use cases that all recipients were classified into: residential quarter, semi-public space, parking lots and park and ride, private on-site parking, nursing services, tourism, and pedelecs. Use-case-specific analysis of the survey results might lead to more intriguing results; e.g., comparing only nursing service or homeowners' associations with each other.

The application of current limit values for nitrogen oxides is based on air pollution data from only a few measurement sites within cities, yet the measured data are utilized by politics for statements concerning whole municipalities. Increasing the number of certified measurement sites within cities affected by high nitrogen oxide values would support the utilization of the correct referent figures for legal measures and increase the foundation of data for area-wide modeling of nitrogen oxide values. Incorporating additional different, yet comparable spatial and traffic settings with additional measurement sites within several cities greatly improves comparative studies between several cities. Addressing this should be a priority moving forward, especially because the World Health Organization plans to rework its recommendations concerning nitrogen oxides to lower the limit values. For a small part, nitrogen oxide values will decline on their own due to newer and cleaner emission concepts and an increasing share of electric vehicles.

The original task to quantify the reduction of nitrogen oxide measurement sites to prevent cities from adopting legal measures for their municipality is not possible for various reasons. Selective measurement sites on certain streets are utilized for statements pertaining to whole municipalities, whereas the specific driving patterns of electric cars are unknown. Deriving spatial correlations between nitrogen oxides, charging infrastructure and electric cars requires enormous amounts of data. Yet, the benefit remains unknown. Calculating the success of charging infrastructure regarding nitrogen oxide reductions utilizing charged electricity data is an appropriate way. In the case of LINOx BW, added-up results for the whole research project in comparison to overall German values seem to be a better way to quantify the success of the project. Nevertheless, the funding of 2.358 charging points in various real-life settings remains a success.

**Author Contributions:** Conceptualization, K.H. and A.G.; methodology, K.H.; software, K.H.; validation, K.H.; formal analysis, K.H. and A.G.; investigation, K.H. and A.G.; resources, K.H. and A.G.; data curation, K.H. and A.G.; writing—original draft preparation, K.H.; writing—review and editing, K.H. and A.G.; visualization, K.H.; supervision, K.H.; project administration, K.H. and A.G.; funding acquisition, K.H. All authors have read and agreed to the published version of the manuscript.

**Funding:** This research was funded by the German Federal Ministry for Economic Affairs and Climate Action (BMWK) (Sofortprogramm "Saubere Luft 2017–2020"); support code 01MZ18012A-D (LINOx BW).

**Data Availability Statement:** Survey data is unavailable due to privacy and funding guidelines within LINOx BW.

**Acknowledgments:** The authors thank the project partners of LINOx BW: the Association of Cities Baden-Württemberg (Städtetag Baden-Württemberg), the Verband Region Stuttgart, the State Agency for New Mobility Solutions and Automotive (e-mobil BW) and the Zentrum für Sonnenenergie- und Wasserstoff-Forschung Baden-Württemberg (ZSW). Furthermore, we would like to thank everyone who participated in phone interviews and the quantitative surveys and thus helped us to gain insight into their respective institutions.

**Conflicts of Interest:** Karsten Hager and Alexandra Graf are employees of Institut Stadt | Mobiltät | Energie (ISME) GmbH. The paper reflects the views of the scientists, and not the company.

## Abbreviations

| | |
|---|---|
| AC | Alternate Current |
| BEV | Battery Electric Vehicle |
| BMWK | German Federal Ministry for Economic Affairs and Climate Action |
| DC | Direct Current |
| EV | Electric Vehicles |
| FCEV | Fuel Cell Electric Vehicle |
| HBEFA | Handbuch für Emissionsfaktoren |
| ICEV | Internal Combustion Engine Vehicle |
| LINOx BW | Ladeinfrastruktur in NOx-Kommunen in Baden-Württemberg |
| PHEV | Plug-In Hybrid Electric Vehicle |
| WTW | Well-to-wheel |

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
