# Peer review of "The Impact of Charging Infrastructure on Local Emissions of Nitrogen Oxides"

_wevj, doi:10.3390/wevj14040090_

Round 1
Reviewer 1 Report
In the manuscript is presented the impact of charging infrastructure on local emissions of nitrogen oxides. 2358 charging within 178 different sub-projects in 23 different cities, over a four year period, were observed and analyzed.
My comments regarding the manuscript are as follows:
- expand the reference list;
- expand the Abstract by referring more to the context of the emissions reduction and electric vehicles;
- expand the Introduction by referring to the work of other researchers;
- add more information regarding the charging points (what is their power, are they regular or fast charging stations and so on);
- the valid cases for reduction research concept (2) are very small (14,3% of the charging points); what is the reason for this?
- present, if the data are available, if there are any differences between the cities (for example if the emissions are lower for Stuttgart or Ludwigsburg);
- present, if the data are available, how the results evolved during the four year period;
- compare the results of this project with other projects or researches that were performed in different countries;
- add a Conclusion Section.
Author Response
Thanks for your remarks, please see the attachment for our replys to your review report.

Reviewer 2 Report
Before publication, the following items should be considered by the author:
1. English must be improved.
2. The novelty of the paper and the new approach to the topic should be highlighted.
3. A list of abbreviations and acronyms should be prepared.
4. The problem statement should be developed.
5. The Journal’s standards for referencing must be considered.
6. To indicate the importance of low-emission technologies and their essential role in the future, the following papers are suggested to be considered:
"Environmental and economic comparison of hydrogen fuel cell and battery electric vehicles." Future Technology 1.2 (2022): 25-33. DOI: 10.55670/fpll.futech.1.2.3
"A comparative study on the energy flow of a conventional gasoline-powered vehicle and a new dual clutch parallel-series plug-in hybrid electric vehicle under NEDC." Energy Conversion and Management 218 (2020): 113019.
"Alternatives to lithium-ion batteries in electric vehicles." Future Technology 1.1 (2022): 33-34. DOI: 10.55670/fpll.futech.1.1.5
7. Clarify how do the charging point installation numbers funded by LINOx BW compare with the overall installation numbers in Baden-Württemberg, and what factors could be responsible for the difference in the kW/charging point ratio?
8. Explain the challenges that the researchers faced in collecting data for the study, particularly with regard to the respondents' familiarity with research projects and the availability of data in their organizations.
9. What are the limitations of the current analysis in quantifying the emission reduction potential of the installed charging infrastructure, and what additional data is required to accurately calculate the potential reduction in emissions?
10. Explain the potential benefits of increasing the number of certified measurement-sites within cities affected by high nitrogen oxide values.
11. The paper needs a conclusion.
Author Response

(The authors gave the same response as above.)

Reviewer 3 Report
Almost all companies that produce electric vehicles now develop or produce chargers. This is very important for effective electrical transport and favorable impact on ecology. In this regard, the issue of properly designing charging infrastructure is very important. However, a little thought and commented on the influence of the charger infrastructure scheme of local emissions of pollutants, in particular from particulate matter and nitrogen oxides. In this regard, the study presented by the authors is very current and useful.
Some remarks
1. References cannot be cited in block (eg. [3-8]-[9-13]). For each of them, describe which aspects are significant for your paper.
2. In line 87 to adjust the literature - [22: 17].
3. I suggest that the discussions be revised and support the conclusions with some quantitative data.
4. It is not clear how the Euro certificate of cars affects the studies presented.
5. I recommend making a thorough analysis of the information in Table 5 and to draw the relevant conclusions.
Author Response

(The authors gave the same response as above.)

Round 2
Reviewer 2 Report
Accept